# The Role of ENHO in Pancreatic Adenocarcinoma: A Bioinformatics Approach [note 1]

**DOI:** 10.3390/cancers17132139

**Published:** 2025-06-25

**Authors:** Osama M. Younis, Zeid K. Al-Sharif, Ahmad E. Saeed, Fares B. Qubbaj, Jehad A. Yasin, Tasnim Nour, Yassine Alami Idrissi, Anwaar Saeed

**Affiliations:** 1Division of Hematology & Oncology, Department of Medicine, University of Pittsburgh Medical Center (UPMC), Pittsburgh, PA 15232, USA; osamayns301@gmail.com (O.M.Y.); yalamiidr@gmail.com (Y.A.I.); 2School of Medicine, The University of Jordan, Amman 11942, Jordan; zaidalsharif123@gmail.com (Z.K.A.-S.); fares.baha@hotmail.com (F.B.Q.); jehadamerjehadyasin@gmail.com (J.A.Y.); t.nour@ju.edu.jo (T.N.)

**Keywords:** ENHO, pancreatic adenocarcinoma, Adropin, bioinformatics

## Abstract

Pancreatic adenocarcinoma is an aggressive cancer that has a 5-year overall survival rate of 13%. *ENHO* encodes for Adropin, a protein that is known to modulate glucose and fat metabolism. Previous studies have shown that exogenous Adropin administration to pancreatic cancer cell lines results in cancer growth. In this study, we sought to study the role of endogenous *ENHO* expression by assessing the transcriptomic landscape associated with *ENHO* expression in pancreatic adenocarcinoma. We identified that increased *ENHO* expression was associated with better overall survival. Further exploration of this discrepancy revealed that *ENHO* expression was associated with a protective immune microenvironment, protective noncoding RNA expression profile, and a microenvironment with downregulated extracellular matrix reorganization and epithelial-mesenchymal transition. We propose *ENHO* as a protective prognostic biomarker in pancreatic adenocarcinoma.

## 1. Introduction

Pancreatic adenocarcinoma (PAAD) is an aggressive malignancy by all standards. It is the third leading cause of cancer-related deaths amongst males and females in 2024 in the United States [1], and some estimate it will soon become the second leading cause by 2030 [2]. PAAD is notoriously linked with a poor prognosis, highlighted by a five-year survival rate of roughly 13% [3]. This underscores the urgent need for comprehensive investigations into the genetic underpinnings of its pathogenesis and the prognostic implications.

One genetic factor of emerging interest is the energy-homeostasis-associated (*ENHO*) gene, which plays a vital role in lipid and glucose metabolism. Experimental studies indicate that *ENHO* is regulated by the nuclear Liver X Receptor (LXR), a sensor that adjusts its activity in response to lipid and glucose levels. Specifically, when high fat levels are detected, *ENHO* expression is upregulated, and its protein product, Adropin, subsequently facilitates fat storage through the modulation of hepatic lipogenic proteins and interaction with the Peroxisome Proliferator-Activated Receptor Gamma (PPAR-γ), a key mediator of adipogenesis [4]. Additionally, Adropin has shown its influence on endothelial cell function and promotes angiogenesis by stimulating Endothelial Nitric Oxide Synthase (eNOS) expression via the VEGFR2-PI3K-Akt and VEGFR2-ERK1/2 signaling pathways [5].

PAAD, like other pancreatic cancers, sustains its growth and survival through three metabolic adaptations. These include reprogramming intracellular metabolism for glucose, amino acids, and lipids; intricate cross-talk with components of the tumor microenvironment; and enhanced nutrient acquisition through scavenging and recycling mechanisms [2].

The binding between Adropin and its putative receptor, GPR19, further highlights the complex role of this pathway in cancer biology. In mesenchymal-like breast cancer, for example, overexpression of GPR19 stimulates E-Cadherin expression and induces an epithelial-like phenotype via the MAPK/ERK1/2 pathway, thereby contributing to carcinogenesis and metastasis. Paradoxically, other studies have reported that high-level Adropin can induce apoptosis in an MCF-7 breast cancer cell line [6]. Similarly, in colorectal cancer, decreased Adropin expression has been linked to disease progression, with the carcinoma cell expression of Adropin inversely correlated with macrophage infiltration. Moreover, high concentrations of Adropin appear to promote pro-tumorigenic activities by inhibiting inflammasome activation and favoring the induction of M2 macrophages. Conversely, the upregulation of Adropin in tumor-associated macrophages correlates with increased metastasis and invasion. At the same time, transfection of the *ENHO* gene into MC38 colon cancer cells has shown inhibition of tumor growth in vivo. Moreover, low-dose Adropin treatment may enhance antitumor functions by stimulating inflammasome activity and promoting M1 macrophage induction [7].

In the context of pancreatic ductal adenocarcinoma (PDA), the treatment of xenografts with Adropin has been observed to promote cancer cells’ proliferation and migration, concomitant with the upregulation of P-VEGFR2, Ki67, Cyclin D1, and Matrix Metalloprotein 2 expression. In contrast, the knockdown of Adropin yields opposite effects, effectively limiting PDA growth both in vitro and in vivo [8].

Given *ENHO*’s gene’s central role in regulating energy homeostasis and, by extension, the metabolic adaptations critical for PAAD survival, there is compelling evidence for investigating its clinical implications. In light of the paradoxical effects observed for *ENHO* and its product Adropin across different cancer types, this study aimed to delineate the genetic and transcriptomic landscape of *ENHO* in PAAD patients, elucidating its prognostic significance and association with the tumor microenvironment.

## 2. Methods

### 2.1. Patient Population

The Cancer Genome Atlas (TCGA) PAAD project (https://portal.gdc.cancer.gov/projects/TCGA-PAAD, accessed on 28 April 2024) contains 187 sequenced pancreatic adenocarcinoma tumor samples at different stages of disease. This included 183 patients with *ENHO* expression counts. We retrieved the mRNA and miRNA raw count data for all patients. All subsequent analyses were performed either directly on this dataset or via online webtools that utilize the same patient cohort.

### 2.2. Differential Gene and miRNA Expression

First, we compared *ENHO* expression in normal pancreatic tissue with that in pancreatic cancer tissue using TNMnet (https://tnmplot.com/analysis/, accessed on 28 April 2024) [9]. Further differential gene expression analysis based on *ENHO* expression was carried out by labeling the top and bottom 25% of *ENHO* gene expressors in the TCGA PAAD cohort as “high” and “low” expressors, respectively. Differential expression analysis for both genes and miRNA between these groups was conducted using the “DeSeq2” package in R v4.3.2 [10]. For this analysis, we considered a Log2 fold change greater than 0.25 and an FDR-adjusted *p*-value below 0.05 as statistically significant. Lastly, the 25% cut-off was used to provide an adequate number of samples while maintaining enough variability in *ENHO* expression to represent biologically meaningful differences.

### 2.3. Survival Analysis

To evaluate the prognostic significance of *ENHO* in PAAD, survival analysis was conducted using different statistical models. A univariate Cox proportional hazards model was generated using the R packages “survival” [11] and “survminer” [12], with optimal cut-off points defined for *ENHO* expression. In addition, multivariate Cox regression analysis was conducted using TIMER2.0 (http://timer.cistrome.org, accessed on 28 April 2024) [13] for the same cohort. To further strengthen the prognostic significance of *ENHO* findings, Gene Set Variation Analysis (GSVA) based survival analysis using both upregulated and downregulated gene sets was retrieved from GSCAlite (https://guolab.wchscu.cn/GSCA/#/, accessed on 28 April 2024) [14].

### 2.4. Mutation Analysis

The mutational landscape of *ENHO* was characterized by gene gains, amplifications, shallow deletions, and deep deletions. We also explored the correlation between *ENHO* expression and tumor mutational burden and metaploidy. These analyses were performed using TCGEx [15] and cBioportal [16,17,18].

### 2.5. Drug Sensitivity

To assess how *ENHO* expression might influence drug response in PAAD, we conducted a drug sensitivity analysis using GSCAlite. This tool utilizes the Cancer Therapeutics Response Portal (CTRP) and the Genomics of Drug Sensitivity in Cancer (GDSC) datasets [14], both of which are large consortia of drug perturbation data in cancer cell lines. Correlation between gene expression and drug resistance was retrieved and represented via bubble plots. A positive correlation indicates that increased gene expression is related to drug resistance, while a negative correlation indicates drug sensitivity.

### 2.6. Immune Infiltration

The relationship between *ENHO* expression and different immune cells’ infiltration in PAAD was examined through the immune infiltration analysis. Immune infiltration data were obtained from GSCAlite [14], which uses the “ImmuCellAI” algorithm [19] to quantify the expression of 24 distinct immune cell types. Furthermore, correlations between *ENHO* and other immune cell types were analyzed using the CIBERSORT ABS algorithm provided by TIMER2.0 [13]. Lastly, the prognostic impact of these immune cells was evaluated using data from TCGEx [15].

### 2.7. Protein–Protein Interactions and Co-Expressed Genes

A protein–protein interaction (PPI) network was generated using STRING (https://string-db.org/, accessed on 28 April 2024) [20] to identify proteins interacting with *ENHO*. Additionally, co-expressed genes were extracted using cBioPortal [16,17,18], with a focus on correlations between *ENHO* and key chemokines, human leukocyte antigens (HLAs), immune checkpoints, and immunostimulatory genes, and visualized. These genes were specifically chosen in order to show how *ENHO* correlates with the immune environment within a bulk RNA dataset.

### 2.8. Gene Set Enrichment Analysis and Transcription Factor Activity

Gene set enrichment analysis (GSEA) was conducted on both upregulated and downregulated genes using Enrichr [21,22,23] to identify biological pathways associated with *ENHO* expression in PAAD. The top 10 enriched terms from Gene Ontology (GO) (biological processes, molecular functions, and cellular components), Reactome, MsigDB, and KEGG were extracted and visualized using the ggplot2 package in R v4.3.2. Lastly, we analyzed transcription factor (TF) expression and activity using the NetAct R package. This analysis was based on the top and bottom 10% of *ENHO* expressors in the TCGA PAAD, with the NetAct package quantifying the TFs’ expression, and subsequently using differentially expressed TF targets to quantify their activation [24]. The top and bottom 10% of *ENHO* expressors were chosen to provide a balance between the number of patients and meaningful results. Specifically, the NetAct package was made to compare differences between wet-lab experiments, which usually have a lower number of samples but higher biological variation. Using the top and bottom 10% of expressors was able to impose similar results.

### 2.9. mRNA-miRNA Interactions

Validated interactions between differentially expressed miRNA and downregulated mRNA were retrieved from mirTarBase and Targetscan using the “MultiMiR” package in R v4.3.2. A network was then generated using CytoScape along with its Cytohubba add-on to assess network metrics. Furthermore, the correlation between miRNA and mRNA expression was validated in the TCGA PAAD cohort using TCGEx and LinkedOmics [15,25].

## 3. Results

### 3.1. Comprehensive Analysis of ENHO Gene Alterations, Expression, and Clinical Correlations

The study population comprised a total of 183 pancreatic cancer patients with available *ENHO* expression data, stratified into groups based on median (Med) and quartile (Q1, Q3) cutoffs for STAR count expression levels. The mean age at index was 65.01 years (SD ± 11.16), with patients in the lower median group significantly older than those in the higher median group (66.45 vs. 62.85 years, *p* = 0.0287). Consistently, patients in the lower quartile (≤Q1) were older than those in the upper quartile (≥Q3) (66.00 vs. 61.56 years, *p* = 0.0411). The cohort included 82 females (44.8%) and 101 males (55.2%). Race distribution was predominantly White (n = 161, 88%), followed by Asian (n = 12, 6.6%) and Black or African American (n = 6, 3.3%). Prior malignancy was reported in 19 patients (10.4%), and 105 participants (57.4%) reported a history of alcohol use.

Disease types were mainly ductal and lobular adenocarcinomas (n = 145, 79.2%), with a smaller proportion of adenomas/adenocarcinomas (n = 32, 17.5%) that were not further classified into ductal adenocarcinomas or other subtypes. The primary diagnosis had a significant association with *ENHO* expression groups *(*≥Med or <Med) (*p* = 0.012). Most tumors originated in the head of the pancreas (n = 131, 71.6%), and most samples were from primary tumor sites (n = 178, 97.3%), in contrast with only one sample (0.50%), which was metastatic.

Pathologic staging showed a predominance of stage IIB (n = 122, 66.7%) and T3 tumors (n = 147, 80.3%). Node involvement (N1) was common (n = 123, 67.2%), while distant metastases (M1) were rare (n = 5, 2.7%) (Appendix A).

Smoking history varied, with a mean of 1.43 cigarettes/day and 23.39 years smoked. The tumor dimensions and weight at sampling showed large variation. Markedly, overall survival time differed significantly between expression groups, favoring higher *ENHO* expression: patients in the ≥Med group had a longer mean overall survival (667.53 vs. 454.45 days, Wilcoxon *p* = 0.0006), as did those in the ≥Q3 group compared to ≤Q1 (785.73 vs. 472.94 days, Wilcoxon *p* = 0.0011) (Appendix A).

The prevalence of *ENHO* alteration was examined across samples, showing a high frequency of mRNA expression and gene amplification, while only a few samples showed downregulation of *ENHO,* as shown in Figure 1a,b. Despite these variations, it was found that *ENHO* is consistently downregulated in PAAD relative to normal pancreatic tissue (Figure 1c). Furthermore, a negative correlation was observed between *ENHO* expression and both mutation count (rho = −0.17, *p* = 0.0434) and tumor ploidy (rho = −0.17, *p* = 0.038), whereas no significant correlation between *ENHO* expression and TMB (rho = 0.03, *p* = 0.767) was found (Figure 1d–f).

### 3.2. ENHO Expression in Different Survival Outcomes

In order to assess the clinical relevance of *ENHO* expression in PAAD patients, extensive survival analysis was performed across four survival metrics: overall survival (OS), progression-free survival (PFS), disease-free interval (DFI), and disease-specific survival (DSS). Higher *ENHO* expression was consistently associated with better patient outcomes (Figure 2a). Moreover, a multivariate Cox regression model incorporating *ENHO*, Gender, Stage, and Age demonstrated a significant protective effect for *ENHO* (HR = 0.597, 95% CI: 0.419–0.852, *p* < 0.01) (Appendix A). Lastly, the prognostic role of the top 100 downregulated and top 100 upregulated genes based on high and low expression of *ENHO* in the TCGA PAAD cohort was explored. GSVA indicated that the top 100 downregulated genes were associated with poorer outcomes, whereas the upregulated genes exhibited a protective effect across all survival metrics, as illustrated by the bubble plot for OS, PFS, DSS, and DFI shown in Figure 2b,c.

### 3.3. ENHO Expression and Drug Sensitivity

*ENHO*’s relationship to drug sensitivity was explored by utilizing two drug databases that provide information on the sensitivity of cancer cells to different drugs: the Genomics of Drug Sensitivity in Cancer (GDSC) database and Cancer Therapeutics Response Portal (CTRP) database. Figure 3 shows the correlation between *ENHO* expression and drug resistance to 5-Fluorouracil, Paclitaxel, and Gemcitabine, which are all drugs that are commonly used to treat pancreatic adenocarcinoma. The CTRP database showed that *ENHO* was not significantly associated with resistance to any of the three drugs. However, the GDSC database showed that ENHO was positively correlated with resistance to Paclitaxel (r = 0.17, *p* = 0.03) and Gemcitabine (r = 0.13, *p* = 0.002).

### 3.4. Immune Cell Infiltration

Given that PAAD is widely recognized as an immune desert, the role of *ENHO* in immune infiltration was examined. First, a correlation analysis was conducted between *ENHO* expression and the CIBERSORT absolute estimation of infiltration for nine different immune cell types. Increased *ENHO* expression was significantly correlated with higher infiltration of CD8+ T cells (rho = 0.345, *p* < 0.001), memory resting CD4+ T cells (rho = 0.377, *p* < 0.001), M2 macrophages (rho = 0.275, *p* < 0.001), and plasma B cells (rho = 0.23, *p* = 0.0024). Conversely, a significant negative correlation was observed between *ENHO* expression and the infiltration of M0 macrophages (rho = −0.283, *p* < 0.001) and resting NK cells (rho = −0.155, *p* = 0.043). Correlations with M1 macrophages, neutrophils, and activated CD4 memory cells did not reach statistical significance (Figure 4a).

To further strengthen our findings, an immune infiltration analysis was performed using the ImmuCellAI algorithm. The results were consistent with the previous findings, demonstrating positive correlations between *ENHO* expression and the infiltration of CD4+ T cells and CD8+ T cells, in addition to new associations with mucosal-associated invariant T cells (MAIT), NK cells, T follicular helper cells, and gamma delta T cells. Moreover, increased *ENHO* expression was associated with decreased infiltration of regulatory T cells (nTreg), monocytes, and dendritic cells (Figure 4b). The prognostic impact of each immune cell type in PAAD was also evaluated; however, none of the immune cell populations associated with *ENHO* expression had any significant prognostic effect (Figure 4c).

Finally, a correlation analysis was conducted to assess the association between *ENHO* expression and key immunological molecules, including chemokines, human leukocyte antigen (HLA), immune checkpoint inhibitors, and immunostimulatory genes. *ENHO* expression was positively co-expressed with chemokines such as CCL2, CCL4, CCL5, CCL14, CCL16, CCL19, CCL21, XCL2, and CXCL12 while being negative co-expression was noted with CCL20 and CXCL5. With regard to immunostimulatory molecules, positive co-expression was observed with CD27, CD40LG, IL6R, KLRG1, TNFRSF17, and CD48, whereas negative co-expression was detected with CD276 (B7-H3) and CD70, MICB and NT5E. In terms of antigen presentation, ENHO was positively co-expressed with HLA-DMA, HLA-DOA, HLA-DPA1, HLA-DPB1, and HLA-DRB1, which are integral to MHC class II-mediated antigen presentation and T cell activation, while negative co-expression with TAP1, TAP2, and TAPBP, which are genes involved in MHC class 1 antigen presentation. Lastly, ENHO was positively co-expressed with the immune checkpoint molecules PD-1 and LAG3 (Figure 5a).

### 3.5. Gene Set Enrichment and Variation Analysis

GSE was performed on both upregulated and downregulated genes to elucidate the biological processes underlying *ENHO* expression patterns. The upregulated genes mainly showed enrichment in pathways associated with cellular homeostasis and neuronal regulation. Terms such as “Neuronal system”, “Neuron projection”, and “Ionotropic glutamate receptor complex” were enriched. In addition, pathways related to potassium and calcium channel activities, as well as those involved in the regulation of insulin and pancreatic secretion, were concomitantly enriched (Figure 5b). We also noted the upregulation of terms related to insulin secretion and pancreatic beta cell function. In contrast, the downregulated gene set was primarily enriched for terms related to extracellular matrix function and regulation. Enriched terms included “collagen formation”, “extracellular structure organization”, “external encapsulating structure organization”, “extracellular matrix organization”, and “collagen biosynthesis and modifying enzymes”. Moreover, processes such as epithelial to mesenchymal transition (EMT), platelet-derived growth factor, hypoxia, and insulin-like growth factor 2 mRNA binding proteins were also enriched (Figure 5c).

Looking at the differentially expressed genes, we saw that genes related to proper phasic insulin secretion were upregulated in patients with higher *ENHO* expression. In particular, there was an upregulation of genes that are crucial to the normal functioning of insulin secretion and sensitivity in the body. These were *GLUT2* (*SLC2A2*), *GLUT4* (*SLC2A4*), and key beta cell ion channels such as *KCNJ11* and *CACNA1D*. Furthermore, we also saw the upregulation of key IGF buffers, such as *IGFBPL1*, *IGF2-AS*, and *IGFALS.*

On the other hand, genes related to the Warburg effect, such as *GLUT1* (*SLC2A1*) and *HK2,* were downregulated. This was accompanied by the downregulation of downstream targets of *HIF-1a*, a key transcription factor up-regulated in the Warburg effect, such as *ANGPTL4*. We also saw the decreased expression of *IGF2BP1*, *2*, *3*, and multiple members of the IGFL family of peptides, all of which act to increase the stability of IGF or mimic its function (Appendix A).

Further investigation into the transcriptional landscape of high *ENHO* expressors was carried out using GSVA. A set of ten well-established cancer pathways was used as a benchmark for this evaluation and assessed the activity of these pathways in three groups of genes: upregulated, downregulated, and co-expressed genes. The analysis revealed that the upregulated genes showed a significant correlation with *TSC/mTOR* activation. In contrast, the downregulated genes were significantly correlated with both *PI3K/AKT* and *TSC/mTOR* pathways. Lastly, co-expressed genes showed a positive correlation with the *TSC/mTOR* and *PI3K/AKT* pathways, while showing a negative correlation with apoptosis (Figure 6b–d).

### 3.6. Transcription Factor Activity

Transcription factor activity analysis was performed in an attempt to uncover the underlying transcriptional network associated with high ENHO expressors. A heatmap (Figure 6a) was generated to display the differential activation of four activated transcription factors between high and low *ENHO* expressor groups. In particular, a cluster of high *ENHO* expressors was characterized by decreased activity of MYC and POU3F1, coupled with increased activity of DDIT3. Lastly, a small subset of five high expressors and two low expressors demonstrated increased activity of ATF6 (Figure 6a).

### 3.7. miRNA Differential Expression and miRNA-mRNA Interaction Networks

To further characterize the transcriptomic landscape of *ENHO* expressors, miRNA differential expression analysis was performed between high and low *ENHO* expressors. A total of 145 miRNAs were identified as significantly differentially expressed (FDR-adjusted *p*-value < 0.05). Subsequently, miRNA-mRNA interactions between differentially expressed miRNAs and genes were identified. In order to better understand miRNA function in the context of *ENHO*, we linked important enriched terms such as KRAS signaling, epithelial to mesenchymal transition, MET promotes cell motility, signaling by MET, and PI3K/AKT signaling pathway and their differentially expressed genes were integrated with the miRNA profiles, resulting in the miRNA-mRNA-function network illustrated in Figure 7.

Notably, two miRNAs stood out as particularly significant: hsa-miR-1179 (log2FC = 3.784, FDR *p* value < 0.001) and hsa-miR-375 (log2FC = 2.905, FDR *p* value < 0.001). Hsa-miR-1179 had validated interactions with *FN1*, a gene involved in EMT and MET signaling. Furthermore, hsa-miR-1179 had a positive correlation with *ENHO* expression (R = 0.57, *p* < 0.0001), while negative correlations were noted with both *FN1* (R = −0.35, *p* < 0.0001) and MET (R = −0.58, *p* < 0.0001) expression (Figure 8d–f). Hsa-miR-1179 also demonstrated a significant favorable prognostic effect (*p* = 0.0035) as shown by Figure 8h. Similarly, hsa-miR-375 was also associated with a favorable prognostic effect (*p* = 0.033) and primarily interacted with two genes: *KCCN4* and *COL12A1*, which are implicated in KRAS signaling and EMT, respectively (Figure 8g). Hsa-miR-375 was also positively correlated with *ENHO* (R = 0.42, *p* < 0.0001), and negatively correlated with *KCCN4* (R = −0.46, *p* < 0.0001) and *COL12A1* (R = −0.48, *p* < 0.0001) (Figure 8a–c).

In addition, hsa-miR-1197 and hsa-miR-592 were found to be positively co-expressed with *ENHO*, and negatively co-expressed with MET and *FN1*, albeit with associations that were less robust compared with the aforementioned miRNAs (Appendix A). A comprehensive list of differentially expressed miRNA and their respective fold change is provided in Appendix A Lastly, protein-protein interaction networks and their overall effect and tie in with ENHO are found in Appendix A.

## 4. Discussion

### 4.1. ENHO Dysregulation in Pancreatic Cancer

PAAD is an aggressive cancer associated with a 5-year survival rate of only around 13%, primarily due to its late-stage detection, a strong metastatic potential, and limited response to standard treatments [26]. Addressing this clinical challenge requires identifying novel biomarkers to enhance prognostication and guide therapeutic strategies. In the present study, we observed that *ENHO* expression was significantly lower in tumor tissues compared with normal pancreatic tissues (*p* = 3.88 × 10^−68^). This marked reduction suggests that *ENHO* dysregulation is a likely contributor to the development and progression of pancreatic cancer.

Further analysis revealed that *ENHO*’s upregulation in advanced stages of PAAD correlates with better overall survival (HR: 0.597, 95% CI: 0.419–0.852, *p* < 0.01), as well as improved PFS, DFI, and DSS. These findings highlight the clinical relevance of *ENHO* as a potential prognostic biomarker that could allow clinicians to stratify PAAD patients based on their survival risk, and is consistent with our previously reported results [27]. Specifically, patients with lower *ENHO* levels might be identified as higher-risk and could benefit from more intensive monitoring and treatment strategies. Conversely, higher *ENHO* levels could indicate a less aggressive disease course, thus aiding clinicians in treatment planning and patient counseling.

Pathway enrichment analysis further supported *ENHO*’s potential tumor-suppressive role in PAAD. High *ENHO* expression was negatively correlated with pro-oncogenic pathways such as PI3K/AKT, suggesting that *ENHO* may hinder tumor growth by suppressing these pathways. Conversely, low *ENHO* expression was associated with the activation of pathways related to epithelial-mesenchymal transition (EMT), extracellular matrix organization, and cancer metastasis. These findings shed light on *ENHO*’s potential as a protective factor in PAAD by potentially reducing invasiveness and metastatic potential.

Additionally, high *ENHO* expression is associated with a unique transcriptional landscape, characterized by the increased activity of *DDIT3*, a gene involved in integrated stress response pathways that code for pro-apoptotic transcription factors [28]. Concurrently, the transcription factors MYC and POU3F1, both known for driving cell proliferation and oncogenic signaling, were downregulated. Additionally, miRNA analysis revealed that hsa-miR-592 positively correlated with *ENHO* expression and was linked with the downregulation of oncogenes such as *MET* (a receptor tyrosine kinase that promotes cell growth and metastasis) and *HMGA2* (a transcriptional regulator involved in tumor progression and EMT). This negative correlation further highlights its potential tumor-suppressive function.

Similarly, the analysis also identified hsa-miR-1179 and hsa-miR-1197 as miRNAs that are positively associated with *ENHO* expression. These miRNAs also exhibited tumor-suppressive roles by negatively correlating with key oncogenes such as *FN1* and *MET*. Notably, both hsa-miR-592 and hsa-miR-1179 were significantly upregulated in high *ENHO* expressors, showing positive fold changes compared to low *ENHO* expressors. Moreover, hsa-miR-375 was also upregulated in high *ENHO* expressors and negatively correlated with *KCCN4* and *COL12A1*, genes linked to KRAS signaling and EMT, respectively. Together, these findings support the notion that *ENHO* expression has a protective role, potentially contributing to a less invasive and metastatic tumor PAAD phenotype. Furthermore, survival analysis revealed that high expression of hsa-miR-1179 (log-rank *p* = 0.0035) and hsa-miR-375 (log-rank *p* = 0.033) was associated with improved overall survival, further underscoring their role in tumor progression. These data collectively suggest that *ENHO*’s regulation of miRNAs might contribute to its potential as a biomarker and a key player in suppressing aggressive tumor behavior in PAAD.

### 4.2. ENHO and Tumor Microenvironment (TME)

The immunomodulatory role of *ENHO* in the PAAD tumor microenvironment was examined. High *ENHO* expression was observed to be positively correlated with immune infiltration, notably with increased levels of CD8+ T cells, CD4+ T cells, plasma B cells, and M2 macrophages. A positive co-expression of *ENHO* with *TNFRSF17*, a B-cell maturation factor, suggests that *ENHO* may facilitate the recruitment of more mature B cells into the tumor microenvironment, thereby contributing to anti-tumor effects. Additionally, *ENHO* was positively co-expressed with MHC class II presentation genes (including *HLA-DMA*, *HLA-DOA*, *HLA-DPA1*, *HLA-DPB1*, and *HLA-DRB1*), supporting its role in promoting anti-tumor immunity. In contrast, *ENHO* exhibited negative co-expression with MHC class I-related genes (*TAP1*, *TAP2*, and *TAPBP*), which may reflect a tumor-driven mechanism aimed at evading cytotoxic T cell-mediated immune surveillance. Moreover, a positive association between *ENHO* expression and chemokines such as *CCL14* and *CCL16* further indicates that ENHO may be associated with immune cell recruitment within PAAD’s microenvironment.

*ENHO* also demonstrated negative correlations with immune checkpoint molecules *CD276* (B7-H3) and *CD70*, both of which are associated with immune suppression and adverse outcomes in cancer [29,30]. The observed downregulation of B7-H3 in conjunction with high *ENHO* levels suggests a diminished capacity for immune evasion, particularly given *B7-H3*’s known role in promoting tumor progression in pancreatic cancer.

Preclinical studies have further underscored the therapeutic significance of these findings, showing enhanced CAR-T efficacy when targeting CD276 and CD70 simultaneously [31]. Additionally, a modest positive co-expression between *ENHO* and *PDCD1* (PD-1) as well as *LAG3* indicates the potential for effective immunotherapy through immune checkpoint inhibition in patients with high *ENHO* expression.

The data also implies that *ENHO* is involved in a dual state of immunomodulation, where immune activation is promoted yet counteracted by tumor escape mechanisms. For instance, the positive correlation between *ENHO* and *CD27*, along with the negative correlation with its ligand *CD70*, suggests a protective immune response driven by *ENHO*, even as tumor strategies, such as upregulating immune checkpoints like PD-1 and *LAG3* or suppression of MHC-I antigen presentation via the downregulation of *TAP1*, *TAP2*, and *TAPBP*, counterbalance this effect. Moreover, *ENHO*’s negative correlations with other immune checkpoint molecules, such as *CD276* (B7-H3) and *NT5E* (CD73), which are implicated in immune escape by suppressing T-cell activity and promoting regulatory T-cell (Treg) responses [29,32], further reinforce its potential anti-tumor role. These observations are consistent with the findings of Li et al. [33], which highlight the contribution of such immune checkpoints to the progression of PAAD.

Although Xiao et al. reported that *TAP1* and *TAP2* are consistently upregulated in interferon-gamma-dominant PAAD subtypes, underscoring their role in MHC-1 antigen loading and presentation to cytotoxic T-cells [34], the analysis emphasized the downregulation of *TAP1* and *TAP2* as a tumor-mediated strategy for immune evasion. Overall, a more proficient adaptive immune response was observed in association with high *ENHO* expression, which appears to coexist with a tumor-driven attenuation of cytotoxic T cell responses due to the dysregulation of MHC-I and immune checkpoint molecules. These findings suggest that ENHO levels may serve as an indicator of patient response to immune checkpoint blockade in PAAD, although further clinical research is warranted to substantiate this potential.

### 4.3. Drug Sensitivity and Clinical Implications

The analysis of drug sensitivity, using data from the GDSC and CTRP databases, revealed an intriguing and somewhat paradoxical relationship between *ENHO* expression and the efficacy of multiple chemotherapeutic agents. It was observed that higher *ENHO* expression was linked to increased resistance to Paclitaxel and Gemcitabine, which together make up the drug regimen GemTaxol. GemTaxol is used in patients with pancreatic adenocarcinoma and is better tolerated than its more aggressive counterpart, the FOLFIRINOX regimen, which is usually reserved for younger, more fit patients. Despite this, several studies have noted that both regimens offer no difference in overall survival benefit [35].

*ENHO*’s association with increased resistance toward both drugs used in GemTaxol could indicate clinical utility as a resistance biomarker; however, more robust biomarker validation studies are needed to confirm this observation.

### 4.4. Consistencies and Discrepancies with Prior Research

A review of the literature on *ENHO* and its genetic product, Adropin, in the context of PAAD reveals that the analysis may both contextualize and extend the findings of earlier studies. Although research in this area remains limited, the study by Hu et al. represents the only prior investigation of *ENHO*’s role in PAAD. Their in vitro experiments demonstrated that elevated Adropin levels in PANC-1 and CFPAC-1 cell lines enhanced proliferation, migration, and the expression of *p-VEGFR2*, *Ki67*, *cyclin D1*, and *MMP2*. Cells were treated with 50 nM recombinant Adropin (amino acids 34–76), a concentration selected based on preliminary dose-response testing, and assessed 96 h post-lentiviral infection. Mechanistically, Adropin’s pro-tumorigenic effects were mediated via sustained activation of the *VEGFR2* signaling pathway, as shown by the reversal of these effects upon treatment with 16 μM Apatinib, a *VEGFR2* inhibitor. In vivo, Hu et al. established cell-derived xenografts (CDX) in nude mice injected subcutaneously with 1 × 10^6^ PANC-1 cells. Daily intraperitoneal administration of 450 nmol/kg recombinant Adropin resulted in significantly accelerated tumor growth, increased microvessel density (CD31+), and elevated *VEGFR2* pathway activation. Conversely, tumors derived from Adropin-knockdown cells showed reduced growth and angiogenesis [8]. In contrast, the present study found that enhanced *ENHO* expression correlated with improved survival outcomes across multiple metrics, suggesting a potentially protective role. This apparent discrepancy highlights the need for further investigation to clarify the context-specific functions of *ENHO* and Adropin in PAAD.

This divergence from the findings of Hu et al. potentially arises from a combination of contributing factors. PANC-1, CFPAC-1 cell lines, and BALB/c nude mouse xenografts lack a functionally mature immune system and may not accurately emulate the complexity of the human tumor microenvironment [36]. This implication limits the applicability of their findings to human immunity, especially considering that *ENHO*’s antitumor effects appear to be mediated by immunomodulation within the tumor microenvironment. In contrast, our analysis captures the natural course of *ENHO* expression in patient tumors, within an intact immunometabolic setting, and examines how endogenous levels correlate with clinical outcomes, rather than testing the effects of recombinant Adropin administration. Furthermore, Hu et al. used supraphysiologic doses of recombinant Adropin (450 nmol/kg), which exceed physiologic serum concentrations (0.222–2.222 nmol/kg) [37]. Adropin’s effects on the immune system are possibly dose dependent, as shown by Jia et al., high doses may promote tumor progression while small doses (<100 ng/mL) amplify antitumor functions [7].

In addition, the literature on *ENHO*’s role in other cancers has produced conflicting findings that appear to be context dependent, varying by cancer type and Adropin dosage. For example, the work of Jia et al. indicated that the overall effect of Adropin effect in MC38 colorectal cancer depends on the administered dose: high doses were shown to promote pro-tumor activities by inhibiting inflammasome activation and inducing M2 macrophages, whereas low doses (<100 ng/mL) enhance antitumor macrophage functions via the stimulation of the inflammasome and induction of M1 macrophage. Furthermore, the upregulation of Adropin in carcinoma cells was negatively associated with macrophage infiltration and positively associated with metastasis and invasion in tumor-associated macrophages. Moreover, the Adropin overexpressing MC38 cells also exhibited increased CD8+ T cell infiltration [7]. These findings are in line with our immune infiltration analysis, where we demonstrated that although *ENHO* expression was associated with increased M2 macrophages, it also correlated with an elevated infiltration of CD8+ T cells, CD4+ T cells, and plasma B cells, thereby indicating a more hostile microenvironment in PAAD patients.

Stelcer et al. reported that exposure to Adropin in HAC15 adrenal carcinoma cell lines stimulates long-lasting proliferation through AKT-mediated pathways, with the proliferative effect being completely abolished by PI3K/AKT inhibitors [38]. In contrast, the pathway analysis presented herein suggested that the gene expression patterns associated with high *ENHO* expression in PAAD were linked to the suppression of PI3K/AKT signaling. Although direct co-expression analysis indicated a positive correlation between *ENHO* and genes involved in PI3K/AKT signaling, the overall data support the notion that while exogenous Adropin may exert pro-tumorigenic effects, endogenous *ENHO* expression plays a more nuanced role in determining patient outcomes.

The study by Tuna et al. found that increased serum Adropin concentrations, induced by chronic or intermittent caloric restriction in an MMTV-TGFa breast cancer mouse model, were associated with a lower incidence of mammary tumors compared to mice with lower serum Adropin levels. Moreover, treatment of MCF-7 and MDA-MB231 breast cancer cells with Adropin significantly reduces cellular proliferation [39]. These findings are consistent with the present study, which found that increased *ENHO* expression was indicative of a less aggressive disease course and improved survival outcomes.

Rao et al. showed that Adropin binding to GPR19 upregulated E-cadherin via the ERK pathway, a process critical for EMT in MCF-7 and MDA-MB-231 breast cancer cells. This mechanism was posited to promote metastasis while also facilitating mesenchymal-epithelial transition, thereby contributing to the homing of primary tumor cells to secondary sites [40]. In this current analysis, upregulation of genes such as *CAV3*, *CAVIN2*, *SGCZ*, *DAG1*, *SNTG1*, and *SNTG2* in high *ENHO* expressors suggested an involvement in maintaining an epithelial phenotype. Yet, contrary to what was presented by Rao et al., it appears that this epithelial phenotype might be protective in PAAD patients, a finding further supported by the downregulation of *OVOL1*, a transcription factor known to drive EMT [41]. These results suggest a pleiotropic relationship between *ENHO* and cancer aggressiveness.

The findings of Nergiz et al., which reported reduced circulating Adropin levels in patients with endometrial cancer, closely parallel the present observation of downregulated *ENHO* expression in PAAD [42]. Both studies underscore the potential diagnostic utility of *ENHO* and Adropin in distinguishing cancer patients from healthy individuals, with the high sensitivity and specificity of Adropin in endometrial cancer reinforcing its role as a clinically relevant biomarker.

The study by Lin et al. demonstrated that the downregulation of miR-1179 in pancreatic cancer promotes cell proliferation through the silencing of E2F transcription factor 5 and that ectopic expression of miR-1179 suppresses migration, invasion, and proliferation by inducing G_0_/G_1_ cell cycle arrest [43]. These findings are consistent with the present, which showed the upregulation of miR-1179 in patients with high ENHO expression. With respect to hsa-miR-592 and has-miR-375-3p, the literature is sparse regarding their role in PAAD; however, the current findings suggest that these miRNAs may exhibit protective effects by targeting genes involved in *MET* signaling and EMT.

Lastly, we hypothesize that *ENHO*’s reflection of an overall favorable tumor microenvironment might come from its direct effect of regulating glucose metabolism and promoting the utilization of glucose in oxidative phosphorylation rather than aerobic glycolysis. The Warburg effect is a phenomenon that occurs when cancer cells shuttle glucose from oxidative phosphorylation to aerobic glycolysis to generate more cellular building blocks integral to their rapid proliferation [44]. Previous literature has shown that Adropin is a known regulator of insulin sensitivity and glucose utilization, where multiple pre-clinical studies have shown increased insulin sensitivity, decreased hepatic gluconeogenesis, and more efficient oxidative phosphorylation when treating multiple diseases such as coronary atherosclerosis and diabetes in mouse models [4]. A similar trend of insulin sensitivity and more physiological control of insulin was apparent in our study. Several upregulated genes, such as *GLUT4*, *GLUT2*, *KCNJ11*, and *CACNA1D,* indicate preserved insulin sensitivity and a more likely phasic secretion of insulin, indicating better insulin regulation. Furthermore, several genes integral to the Warburg effect were downregulated. This included *GLUT1* and *HK2*, which have been previously shown to significantly increase aerobic glycolysis in pancreatic cancer and are induced by *KRAS* and *STAT3*, respectively [45,46]. This, alongside the previous discussion about *ENHO*’s immunomodulatory role, is a plausible explanation for why increased *ENHO* expression is associated with better overall survival in pancreatic adenocarcinoma patients.

### 4.5. Limitations

While this study provides valuable insights into the potential role of *ENHO* as a prognostic biomarker in pancreatic adenocarcinoma (PAAD), several limitations should be noted. First, this bioinformatics-based analysis relied entirely on publicly available datasets, which—despite their comprehensiveness—may not fully capture the heterogeneity of real-world patient populations. Potential biases in sample selection, cohort composition, and data preprocessing could limit the generalizability of the findings.

Second, the study lacks experimental validation of key computational predictions. In particular, the inferred miRNA-mRNA interactions involving *ENHO*, the observed associations with immune infiltration, and its links to oncogenic signaling pathways require confirmation through in vitro and in vivo studies. Such validation is essential for establishing causality, determining *ENHO*’s role in PAAD progression and exploring its therapeutic potential at the transcriptomic and genomic levels.

Third, the use of bulk-RNA sequencing data does not allow for assessment of *ENHO* in specific cell types within the tumor microenvironment, potentially leading to incomplete conclusions. Fourth, no stratified analyses were conducted based on demographic or clinical factors such as age, sex, tumor stage, or comorbidities. These variables could influence *ENHO* expression and function, and should be examined in future research to better assess its utility across diverse patient subgroups.

Fourth, certain hypotheses generated in this study require more substantial evidence. For example, the hypotheses regarding ENHO’s net protective effect against the Warburg effect need to be tested in pre-clinical models. Furthermore, insulin levels and pancreatic function in relation to ENHO need to be tested in vivo and possibly in pancreatic adenocarcinoma patients.

Finally, while comparisons with other cancers provide useful context, discrepancies in the literature highlight the need for further research to clarify *ENHO*’s role across different malignancies. Despite these limitations, the results highlight the translational potential of *ENHO* as a biomarker for patient stratification or therapeutic response prediction in PAAD. Advancing toward clinical application will require rigorous experimental validation and prospective studies to confirm its diagnostic, prognostic, and predictive value.

## 5. Conclusions

The discrepancies between the literature and the present analysis indicated that exogenous Adropin and endogenous *ENHO* expression may not overlap functionally in PAAD. Although Adropin, as the genetic product of the *ENHO* gene, appears to have an overall pro-tumorigenic effect when administered exogenously, by activating VEGFR2 and PI3K/AKT signaling pathways, the upregulation of ENHO in situ appears to confer a net protective effect in pancreatic adenocarcinoma. This protective effect is likely due to *ENHO*’s role in immune modulation, wherein it promotes immune infiltration and fosters a tumor microenvironment that is hostile to tumor growth. Furthermore, the metabolic reprogramming that *ENHO* is known for might play a role in modulating the metabolic utility of glucose in the context of cancer, favoring oxidative phosphorylation over aerobic glycolysis. It is further hypothesized that the protective network associated with *ENHO* upregulation, as evidenced by favorable immune signatures and miRNA interactions, cannot be fully recapitulated by exogenous Adropin treatment.

In summary, it is speculated that serum Adropin levels, which reflect *ENHO* expression within the tumor microenvironment, may more accurately represent the true role of *ENHO* in cancer, particularly in PAAD patients. Further research, through preclinical studies utilizing cancer models that better represent the tumor microenvironment and clinical studies assessing the levels of serum Adropin in pancreatic cancer patients, is warranted to confirm these findings and explore the relationship between serum Adropin levels, tissue *ENHO* expression, and patient outcomes.

## Figures and Tables

**Figure 1 cancers-17-02139-f001:**
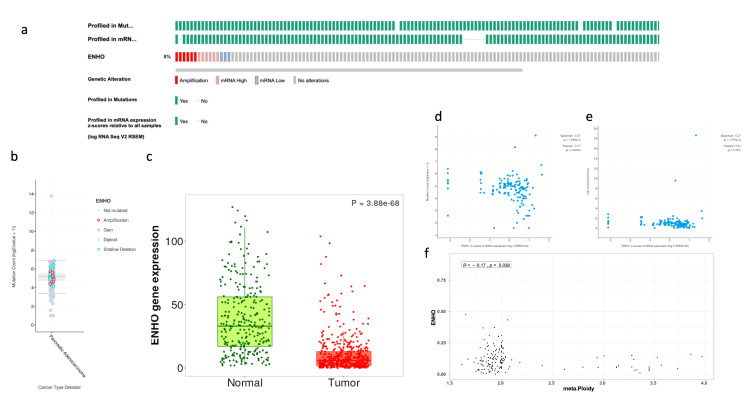
Complete mutational and expression landscape of *ENHO* in pancreatic adenocarcinoma. (**a**) shows onco-print showing the frequency of genetic alterations of *ENHO* in PAAD patients. (**b**) shows a further detailed box plot showing the types of mutations plotted against the mutation count. (**c**) shows the expression of *ENHO* in tumor vs. normal pancreatic tissue. (**d**) shows the correlation between *ENHO* mRNA expression and mutation count. (**e**) shows the correlation between *ENHO* mRNA expression and TMB. (**f**) shows the correlation between ENHO mRNA expression and aneuploidy of PAAD samples.

**Figure 2 cancers-17-02139-f002:**
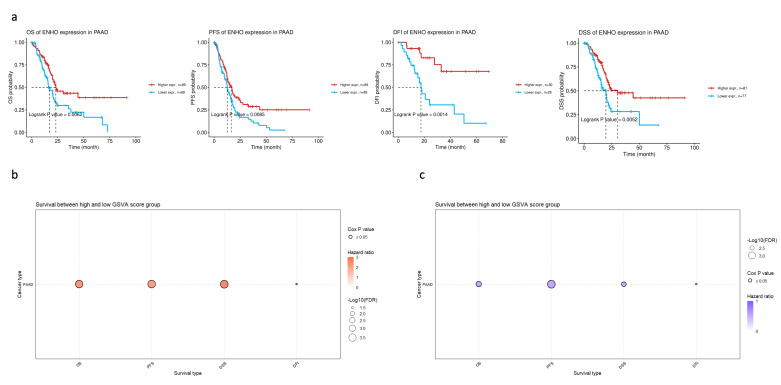
Prognostic role of *ENHO* in PAAD. (**a**) shows Kaplan–Meier plots splitting PAAD patients by their median *ENHO* expression into high and low groups for OS, PFS, DFI, and DSS. (**b**) shows the univariate cox-regression of the top 100 downregulated genes between high and low *ENHO* expressors. (**c**) shows the univariate Cox-regression of the top 100 upregulated genes between high and low *ENHO* expressors.

**Figure 3 cancers-17-02139-f003:**
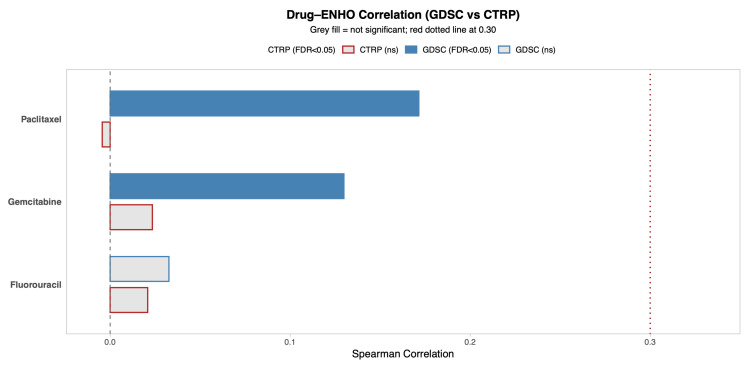
A comprehensive overview of the correlation between *ENHO* expression and drug sensitivity using both the GDSC and CTRP databases. Only drugs relevant to PAAD were included.

**Figure 4 cancers-17-02139-f004:**
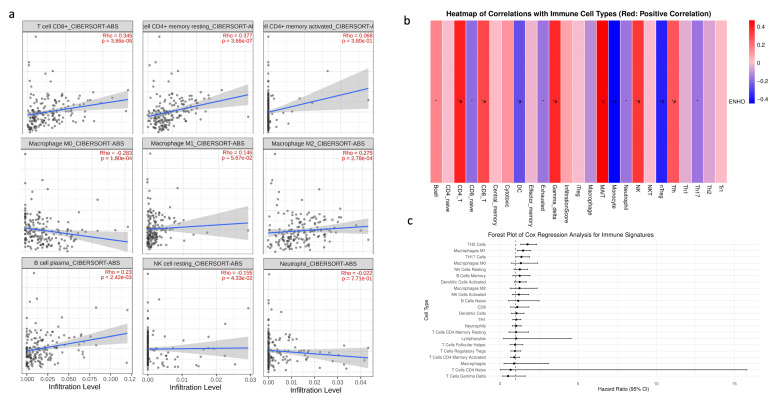
*ENHO* mRNA expression and immune cell infiltration analysis. (**a**) shows the infiltration of CD8+, resting CD4+, activated CD4+ T cells, M0, M1, and M2 macrophages, plasma B cells, resting NK cells, and neutrophils via the CIBERSORT ABS algorithm. (**b**) shows the infiltration of 24 different immune cell types by the ImmuCellAI algorithm. (**c**) shows a forest plot showing a multivariate Cox regression of all the immune cell types in pancreatic adenocarcinoma. *  =  *p* <  0.05, # =  FDR <  0.05.

**Figure 5 cancers-17-02139-f005:**
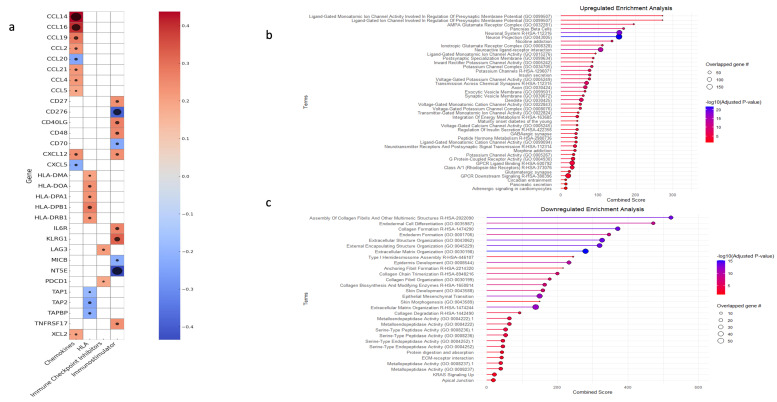
An elucidation of *ENHO*’s immune role, alongside a gene set enrichment analysis for the differentially expressed upregulated and downregulated genes. (**a**) shows a heatmap of the correlations between *ENHO* expression and different chemokines, human leukocyte antigens (HLAs), immune checkpoint inhibitors, and immunostimulators. (**b**) GSEA of differentially expressed upregulated genes. (**c**) GSEA of differentially expressed downregulated genes. The black circles in (**a**) correlate with -log10(FDR), all FDR’s are < 0.05.

**Figure 6 cancers-17-02139-f006:**
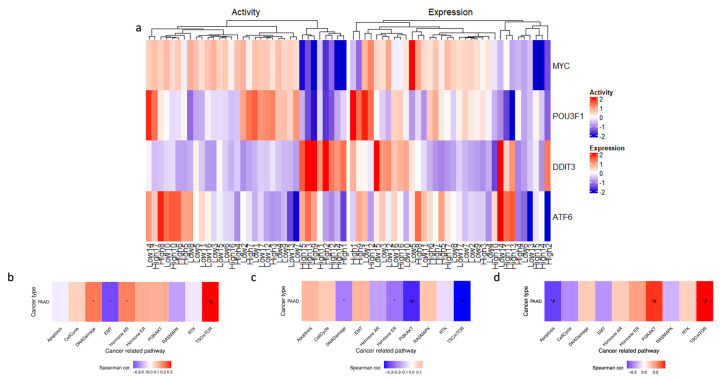
Exploring *ENHO*’s transcriptional landscape through transcription factor activity alongside Gene Set Variation Analysis of upregulated, downregulated, and co-expressed genes. (**a**) shows a heatmap that elucidates differential transcription factor expression and activity between the top and bottom 10% of *ENHO* expressors. (**b**–**d**) show the GSVA in 10 cancer-related pathways for differentially expressed (**b**) upregulated genes, (**c**) downregulated genes, and (**d**) co-expressed genes. *  =  *p* <  0.05, # =  FDR <  0.05.

**Figure 7 cancers-17-02139-f007:**
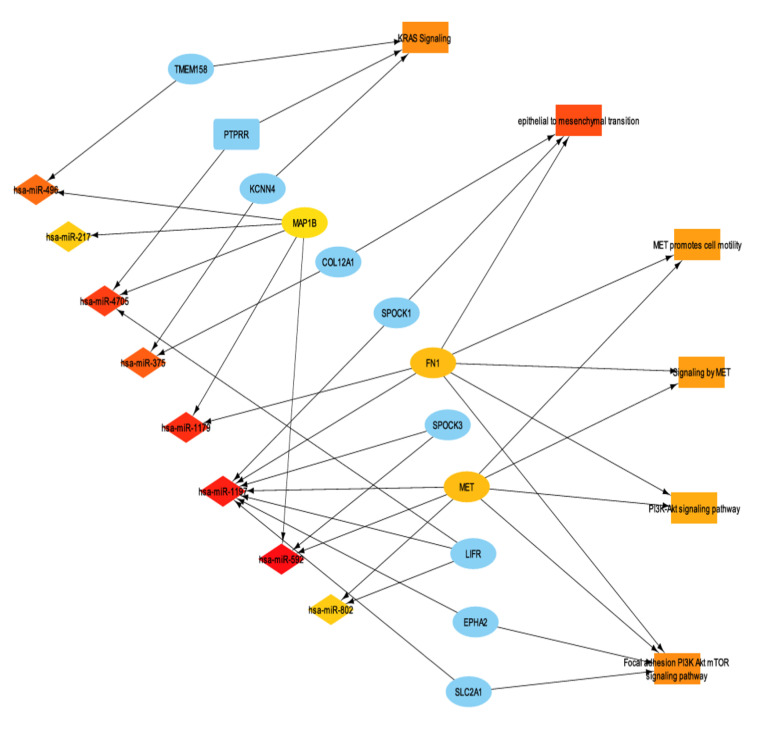
mRNA-mRNA interaction network with functionally enriched terms. It shows a subset of the network that contains the top 30 most connected nodes via maximum clique centrality (MCC).

**Figure 8 cancers-17-02139-f008:**
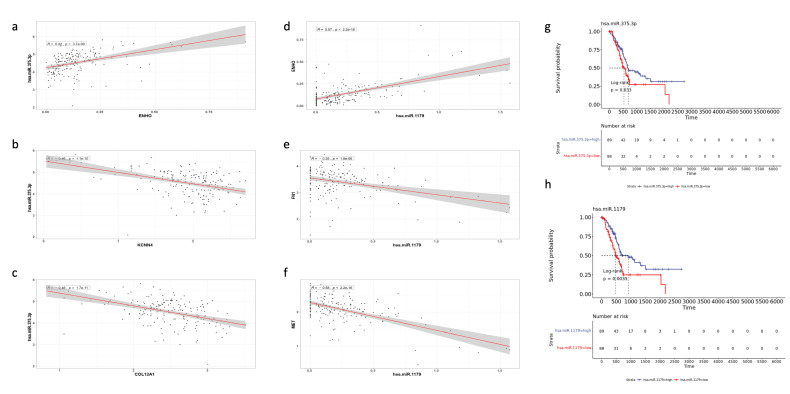
Validation of the role of different miRNAs in PAAD. The figure shows the correlation of MiR-375.3p with (**a**) *ENHO*, (**b**) *KCNN4*, and (**c**) *COL12A1*. Furthermore, the figure shows the correlation between MiR-1179 and (**d**) *ENHO*, (**e**) *FN1*, and (**f**) *MET*. (**g**) Kaplan–Meier of OS for MiR-375.3.p. (**h**) Kaplan–Meier of OS for MiR-1179.

## Data Availability

All the data used in this paper can be accessed publicly via the TCGA at the genomic data commons (https://portal.gdc.cancer.gov/, accessed on 28 April 2024) or through the online analysis webtools mentioned within the methods section of this article.

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
