# Peer review of "The Role of ENHO in Pancreatic Adenocarcinoma: A Bioinformatics Approachâ€"

_cancers, 2025, doi:10.3390/cancers17132139_

Round 1
Reviewer 1 Report
Comments and Suggestions for Authors
This manuscript by Younis et al. presents a comprehensive bioinformatics analysis investigating the role of ENHO in pancreatic adenocarcinoma. The study leverages TCGA data to examine ENHO expression patterns, survival outcomes, immune infiltration, and associated molecular pathways.
The study's major strengths include its multi-faceted approach integrating survival analysis, immune profiling, pathway enrichment, and miRNA-mRNA network analysis. The authors demonstrate convincingly that higher ENHO expression correlates with improved overall survival (HR = 0.597, p < 0.01) and other survival metrics, which contrasts interestingly with prior literature showing pro-tumorigenic effects of exogenous adropin.
However, several limitations warrant attention. The study relies entirely on computational analysis without experimental validation, which limits the ability to establish causality. The use of bulk RNA-seq data prevents cell-type-specific analysis within the tumor microenvironment. The drug sensitivity findings appear paradoxical and somewhat inconclusive, showing both resistance and sensitivity patterns that lack clear interpretation. The discussion of discrepancies with prior literature, particularly the Hu et al. study, could be more thoroughly addressed to explain why endogenous ENHO expression might differ functionally from exogenous adropin administration.
To strengthen this manuscript, the authors may want to consider providing clearer mechanistic hypotheses for how ENHO mediates its protective effects versus the pro-tumorigenic effects of exogenous adropin. The drug sensitivity section needs more coherent interpretation or could be shortened if the findings remain ambiguous. This part is not convincing to this reviewer. Perhaps, including a schematic figure summarizing the proposed protective mechanisms of ENHO would enhance clarity. The limitations section appropriately acknowledges the computational nature of the study. Still, the authors should emphasize the need for experimental validation of key findings, particularly the miRNA-mRNA interactions and immune infiltration patterns. Additionally, discussing potential clinical applications, such as ENHO as a biomarker for patient stratification or therapeutic response prediction, would enhance the manuscript's translational relevance.
Overall, this is a well-executed bioinformatics study that provides valuable insights into ENHO's potential protective role in PAAD. Experimental validation will be essential to confirm these computational findings and establish clinical utility, and this should be discussed in the manuscript.
Author Response
Dear reviewer 1,
First of all, we would like to thank all of you for the time and effort you have put into improving this manuscript, it is greatly appreciated.
Below you will find a detailed response to all of your points with key edits and where to find them.
We are looking forward to your response.
Comment 1:
To strengthen this manuscript, the authors may want to consider providing clearer mechanistic hypotheses for how ENHO mediates its protective effects versus the pro-tumorigenic effects of exogenous adropin. The drug sensitivity section needs more coherent interpretation or could be shortened if the findings remain ambiguous. This part is not convincing to this reviewer. Perhaps, including a schematic figure summarizing the proposed protective mechanisms of ENHO would enhance clarity.
Response 1:
We do agree, the submitted version lacked clear hypotheses as to why ENHO might exert its effect. We have added two main hypotheses that we think drive the findings discussed in our paper. First, the correlation between ENHO expression and CD8+ T cells provides a solid argument as to why increased ENHO expression might be associated with increased survival. This is also supported by the literature where Jia et al showed the MC38 colorectal cancer cell lines overexpressing ENHO also showed increased CD8+ T cell infiltration in mouse models.
Discussion of the immune microenvironment is in lines 512-520
Furthermore, we have added a new hypothesis that is based on the main function of ENHO which is metabolic regulation. We noticed that high ENHO expressors had decreased expression of genes related to the Warburg effect and increased expression of genes responsible for appropriate insulin secretion. This is in line with ENHO’s previous reported role as a regulator of glucose metabolism.
This is discussed in lines 294-306 and lines 565-584.
Lastly, we have shortened the drug resistance section and made it more relevant to pancreatic adenocarcinoma by only including drugs that are used to treat it in clinical practice. This included 5-Fluorouracil, gemcitabine, and paclitaxel. None of the other drugs used to treat PAAD were found in the GDSC or CTRP databases.
This is discussed in lines 229-237 and lines 457-468.
Comment 2:
The limitations section appropriately acknowledges the computational nature of the study. Still, the authors should emphasize the need for experimental validation of key findings, particularly the miRNA-mRNA interactions and immune infiltration patterns. Additionally, discussing potential clinical applications, such as ENHO as a biomarker for patient stratification or therapeutic response prediction, would enhance the manuscript's translational relevance.
Response 2:
We have edited our limitations section to be more inclusive of the aforementioned comments.
Given the new drug resistance results we also discuss how that could shape therapeutic response prediction and treatment selection for these patients.
This is discussed in lines 457-468 and in lines 586-604.
Reviewer 2 Report
Comments and Suggestions for Authors
There are some comments.
It would be beneficial to include pathological characteristics of pancreatic ductal adenocarcinoma, such as histologic grade, histologic subtype, tumor size, and tumor location. If available, please provide information on ENHO expression according to these pathological parameters.
It would be beneficial to provide more detailed descriptions of the methods for mutational and gene analysis to enhance reproducibility and clarity.
It would be better to consider improving the quality and resolution of the Figures. Enlarging the figures would also help readers better visualize the embedded text and data.
Gene names should be written in italics.
Reference duplication has been noted: please check—specifically references 8 and 34, as well as 7 and 35.
Comments on the Quality of English LanguagePlease thoroughly check the manuscript for English grammar and spelling errors.
For example, revelaled ->revealed
bortezomib. conversely, ->bortezomib. Conversely,
Author Response
Dear reviewer 2,
First of all, we would like to thank all of you for the time and effort you have put into improving this manuscript, it is greatly appreciated.
Below you will find a detailed response to all of your points with key edits and where to find them.
We are looking forward to your response.
Comment 1:
It would be beneficial to include pathological characteristics of pancreatic ductal adenocarcinoma, such as histologic grade, histologic subtype, tumor size, and tumor location. If available, please provide information on ENHO expression according to these pathological parameters. It would be beneficial to provide more detailed descriptions of the methods for mutational and gene analysis to enhance reproducibility and clarity.
Response 1:
Please refer to Table S2 in the supplementary material regarding the patient characteristics tables, and lines 179-204.
Comment 2:
It would be better to consider improving the quality and resolution of the Figures. Enlarging the figures would also help readers better visualize the embedded text and data.
Response 2:
All uploaded figures have a 2k resolution, 300 dpi, and are in TIFF format. The quality is high when they are not embedded in the word file. Is there a way we can ensure that it is reflected in the final paper?
Thank you for your efforts!
Comment 3:
Gene names should be written in italics.
Reference duplication has been noted: please check—specifically references 8 and 34, as well as 7 and 35
Response 3:
All gene names are in italics and citations have been sorted out.
Comment 4:
Please thoroughly check the manuscript for English grammar and spelling errors.
Response 4:
The manuscript has been thoroughly checked
Round 2
Reviewer 1 Report
Comments and Suggestions for Authors
Thank you for promptly revising the manuscript and clarifying important aspects of this interesting study.
All concerns and questions have been addressed. I am pleased to inform you that I recommend accepting the manuscript in its present form.
Author Response
Dear Reviewer 1,
Thank you for your recommendation and help in improving our manuscript.
Reviewer 2 Report
Comments and Suggestions for Authors
The manscript was well revised.
There are some comments.
It would be better to define "adenoma", "ductal and lobular neoplasms", "epithelial neoplasms, NOS", and "infiltrating duct carcinoma, NOS" in detail.
It would be better to clarify the photos (Figures 1-8) and the text within them.
Comments on the Quality of English LanguagePlease check English grammar and spelling.
For example, microenvironment, In contrast, -> microenvironment. In contrast,
warburg effect -> Warburg effect
Please unify the use of terminology
For example, Adropin vs. adropin
Author Response
Dear Reviewer 2,
Below you will find a detailed response to all of your points with key edits and where to find them.
We are looking forward to your response.
Comment 1:
It would be better to define "adenoma", "ductal and lobular neoplasms", "epithelial neoplasms, NOS", and "infiltrating duct carcinoma, NOS" in detail.
Response 1:
We did not quite understand what a definition of these subtypes meant. These are just the different types of pancreatic adenocarcinoma that were present in the TCGA-PAAD cohort. A slight elaboration on what ductal and lobular neoplasms are and how they differ from adenomas/adenocarcinomas was made.
Found on line 187-193.
Comment 2:
It would be better to clarify the photos (Figures 1-8) and the text within them.
Response 2:
All Figures 1-8 have been enlarged and their text is now more legible. To do that we removed the captions from the figure directly and instead only relied on the captions at the end of the manuscript.
Comment 3:
Comments on the Quality of English Language
Please check English grammar and spelling.
For example, microenvironment, In contrast, -> microenvironment. In contrast,
warburg effect -> Warburg effect
Please unify the use of terminology
For example, Adropin vs. adropin
Response 3:
We unified the use of terminology and fixed any grammar/capitalization mistakes that we stumbled upon.
Thank you for your continued effort in improving our manuscript! It is much appreciated.